# A Nonlinear Model and Parameter Identification Method for Rubber Isolators under Shock Excitation in Underwater Vehicles

Heye Xiao [1], Chizhen Xu [2], Ruobing Wang [3], Peixun Yu [2,*], Jie Zhou [2] and Junqiang Bai [1]

[1] Unmanned System Research Institute, Northwestern Polytechnical University, Xi'an 710072, China; xiaoheye@nwpu.edu.cn (H.X.); junqiang@nwpu.edu.cn (J.B.)
[2] School of Aeronautics, Northwestern Polytechnical University, Xi'an 710072, China; xuchizhen@mail.nwpu.edu.cn (C.X.); jiezhou@nwpu.edu.cn (J.Z.)
[3] Xi'an Modern Control Technology Research Institute, Xi'an 710065, China; 13891927481@163.com
* Correspondence: yupeixun@nwpu.edu.cn

**Abstract:** Rubber isolators are usually used to protect high-precision equipment of autonomous underwater vehicles (AUVs), avoiding damage from overlarge dynamic excitation. Considering the nonlinear properties of the rubber material, the nonlinear behavior of rubber isolators under shock exaltation is hard to be predict accurately without the available modal and accurate parameters. In view of this, the present study proposes a nonlinear model and parameter identification method of rubber isolators to present their transient responses under shock excitation. First, a nonlinear model of rubber isolators is introduced for simulating their amplitude and frequency-dependent deformation under shock excitation. A corresponding dynamic equation of the isolation system is proposed and analytically solved by the Newmark method and the Newton-arithmetic mean method. Secondly, a multilayer feed-forward neural network (MFFNN) is constructed with the current model to search the parameters, in which the differences between the estimated and tested responses are minimized. The sine-sweep and drop test are planned with MFFNN to build the parameter identification process of rubber isolators. Then, a T-shaped isolator composed of high-damping silicon rubber is selected as a sample, and its parameters were determined by the current identification process. The transient responses of the isolation system are reconstructed by the current mode with the identified parameter, which show good agreement with measured responses. The accuracy of the proposed model and parameter identification method is proved. Finally, the errors between the reconstructed responses and tested responses are analyzed, and the main mode of energy attenuation in the rubber isolator is discussed in order to provide an inside view of the current model.

**Keywords:** rubber isolator; nonlinear behavior; parameter identification method; multilayer feed-forward neural network; fractional-order method; underwater vehicle





## 1. Introduction

Rubber isolators are widely used for the high-precision equipment of autonomous underwater vehicles, air-sea amphibious aircraft, and other underwater vehicles in reducing dynamic loads originating from environmental excitations [1–4],which always include stationary vibration and transient impulse.Compared to the stationary vibrations with limited amplitude and continuous duration, rubber isolators exhibit stronger nonlinear characteristics induced by shock pulse with high amplitude and short duration [5]. Furthermore, the materials used in the isolator are often sensitive to manufacturing processes and may be non-homogenous and anisotropic. Therefore, the transient responses of rubber isolators under shock excitation are more difficult to be obtained by the numerical method merely. The experimental methods must be adopted to identify the parameters of rubber isolators and combined with the numerical model to present its nonlinear characteristics in design processes.

Parameter identification of rubber isolators has become the interest of researchers in the past two decades [6–10]. In the studies, the identification method or process is dependent on the nonlinear model of rubber isolators directly or indirectly. Therefore, the quality of the identified parameters relies on the accuracy of the nonlinear model. In order to address the nonlinear behavior of rubber isolators, various models have been proposed and applied to present the nonlinear behavior of rubber isolators in recent decades, such as fractional derivatives models [11–17], Mats Berg model [18,19], and Maxwell models [20–23]. The fractional derivative models have been used to describe the viscoelastic behavior of elastomeric materials in the isolator with fewer parameters [17] compared to other models, so a small quantity of parameters could decrease the computational work in the identification process. Thus, the fractional derivatives model is an ideal model for the parameter identification of rubber isolators under shock pulses.

Considering shock excitations, the fractional derivatives terms are usually used with stiffness expression to build the nonlinear model of rubber isolators in most research studies [5,15,16,20]. For example, based on the relationship between the mechanism of entropy elasticity and the fractional derivative terms, the nonlinear behavior originating from the fractional derivative, it was analyzed clearly for shock excitations [20]. Furthermore, four different types of nonlinear fractional derivative terms were combined with linear stiffness to build several nonlinear models for investigating their transient response [16]. In this model, the stiffness-related nonlinear behavior failed to be simulated by linear stiffness. To consider the stiffness-related response of the isolator, a dynamic stiffness spectrum was combined with a fractional element to estimate time domain responses of the isolator to a class of transient excitation signals [5]. The nonlinear expressions of irrational stiffness and fractional damping were used to propose a novel isolator for describing various nonlinear stiffness and damping under shock pulse [15]. While the relationship between the mechanism of the material and the nonlinear behavior of the isolator under shock excitation is hardly found in the studies mentioned above. The identified parameters with these models failed to present the mechanical mechanism of the material and failed in helping the material selections in the design.

Therefore, a nonlinear model of rubber isolators under shock pulse, describing the amplitude and frequency depended response of the isolators with the mechanism of the rubber material, is proposed in this paper. Considering the high amplitude of the shock pulse, the relationship between restoring force and the deformation of the rubber material may change from linear to viscoelastic or hysteretic properties in a wide range [24]. To describe the wide range of deformation, the Iwan model [25] and exponential model are combined to describe the friction and the hysteretic deformation of the rubber material in the proposed model. Regrading the wide frequency range related to the short duration of the shock pulse, the fractional derivatives and linear damping are used in the proposed model to present the complex damping effects of the rubber material in low frequency and high frequency ranges, respectively.

Based on the current model, the location and the type of nonlinearity are determined theoretically. Therefore, the current parameter identification of rubber isolators is a problem for parameter estimation of nonlinearities in a classified system [7]. For parameter estimations, there exist several time domain and frequency domain methods in the literature [6–10,26–28]. They use the optimal methods to solve this type of problem efficiently and approached parameters of nonlinear rubber isolator by minimizing the discrepancy between the measured responses and the theoretical responses of the system. Based on this idea, the multilayer feed-forward neural network (MFFNN) theory is used as an optimization tool to determine the parameters of rubber isolators by minimizing the discrepancy between the measured and theoretical responses of the system.The drop tests are used to obtain the measurement data, which are used as baselines of the comparisons for search parameters in MFFNN. In consideration of numerous linear and nonlinear identification parameters, a sine-sweep test is conducted to determine the linear parameters and used as

the initial value to accelerate the parameter searching process and to improve optimization in MFFNN efficiency.

The remainder of this paper is structured as follows. Section 2 introduces a nonlinear model of rubber isolators, which is constructed based on the properties of the rubber material. An MFFNN is constructed for parameter identification with the current model. Then, the entire identification process is explained with the MFFNN and the organization of the sine-sweep and drop tests. In Section 3, the experimental configurations and detailed descriptions of the isolation system sample are presented. The parameter identification results are obtained and used to reconstruct the transient response of the isolation system. Then, the errors between the reconstructed responses and tested responses are analyzed. The energy dissipated by different types of damping and the total dissipation energy is respectively calculated to reveal the main mode of energy attenuation in the isolator. Finally, the conclusions of this research study are provided in Section 4.

## 2. Parameter Identification Method

### 2.1. Nonlinear Dynamic Model of the Isolation System

According to Reference [14], the total restoring force of the nonlinear model is obtained from the direct summation of three forces assigned to each component of the model as follows:

$$F_t(t) = F_e(t) + F_{ve}(t) + F_f(t) \tag{1}$$

where $F_e(t)$, $F_{ve}(t)$, and $F_f(t)$ are the restoring elastic, viscoelastic, and friction forces, respectively. According to the stretching and compression behavior of rubber, the frictional force $F_f(t)$ is modeled by the Iwan model, the expression of which is described in Appendix A.1. Furthermore, the hyper elastic force is simulated by the exponential function, which is presented in detail in Appendix A.2.

Considering wide frequency excitation, this mode of energy attenuation can be described by combining fractional-order and constant-order derivations. The viscoelastic damping force can then be defined as follows:

$$F_{ve} = cDx(t) + bD^\alpha x(t) \tag{2}$$

where $bD^q x(t)$ is the fractional-order derivation, the physical implications and definition of which are explained in Appendix A.3, and $cDx(t)$ is constant-order derivation, and it presents linear damping of the linear oscillator in the Kelvin–Voigt model.

The weight of the high-precision equipment with isolators, such as navigation equipment, is always smaller than the floating body. The protected equipment and supporting rubber isolators can be simplified as a one-degree nonlinear oscillator, as shown in Figure 1. The nonlinear mechanic relationship between restoring forces and deformation of the rubber material is detailed in Figure 1.

Based on the nonlinear mechanic relationship between restoring force and deformation of the rubber material, an isolation system consisting of the protected equipment and supporting rubber isolators can be simplified as a one-degree nonlinear oscillator, as shown in Figure 1. Accordingly, the dynamic equation of the isolation system driven by an acceleration pulse is given by Equation (3):

$$m\ddot{x}(t) + c\left(\dot{x}(t) - \dot{x_0}(t)\right) + b\frac{d^\alpha\left[\frac{x(t)-x_0(t)}{1-\mu(x(t)-x_0(t))}\right]}{d^\alpha t} + F_f(x(t) - x_0(t)) + F_e(x(t) - x_0(t)) = 0 \tag{3}$$

where $m$ is the system mass. If $u(t) = x(t) - x_0(t)$ is set, then the following is the case.

$$mu(\ddot{t}) + cu(\dot{t}) + b\frac{d^\alpha\left[\frac{u(t)}{1-\mu u(t)}\right]}{d^\alpha t} + F_f(u(t)) + F_e(u(t)) = -m\ddot{x}_0(t) \tag{4}$$

Assuming that $a(t) = -mx_0(\ddot{t})$, we have the following case.

$$mu(\ddot{t}) + cu(\dot{t}) + \frac{d^\alpha b\left[\frac{u(t)}{1-\mu u(t)}\right]}{d^\alpha t} + F_f(u(t), \{z_i\}) + F_e(u(t)) = a(t) \tag{5}$$

The Newmark method [29–31] provides a convenient approach for the solution of the second-order linear differential equation, which can express the velocities and accelerations at the next time step by combining the known parameters at the previous time step and the unknown displacement at the next time step. Therefore, it is convenient to adopt the Newmark method to solve Equation (5) when the initial conditions of the velocities and displacements are known. The transformed relationship [31] is given as follows:

$$\begin{cases} \dot{u}(t + \Delta t) = -\frac{\beta}{\lambda\Delta t}u(t) - \left(\frac{\beta}{\lambda} - 1\right)\dot{u}(t) - \left(\frac{\beta}{2\lambda} - 1\right)\Delta t\ddot{u}(t) + \frac{\beta}{\lambda\Delta t}u(t + \Delta t) \\ \ddot{u}(t + \Delta t) = -\frac{\beta}{\lambda\Delta t^2}u(t) - \frac{1}{\lambda\Delta t}\dot{u}(t) - \left(\frac{1}{2\lambda} - 1\right)\ddot{u}(t) + \frac{1}{\lambda\Delta t^2}u(t + \Delta t) \end{cases} \tag{6}$$

where $\Delta t$ is an increment of time, and $\lambda$ and $\beta$ are parameters that can be varied at different time steps. The nonlinear stiffness can be described as a discretized expression according to discrete time steps, and it is given as follows:

$$\begin{cases} F_f[u(t + \Delta t)] = \sum_{i=1}^{l} k_i u(t + \Delta t) + C_k \\ F_e[u(t + \Delta t)] = k_d e^{\frac{u(t+\Delta t)\pi}{2d}} \end{cases} \tag{7}$$

where the following is the case.

$$C_k = \sum_{i=1}^{n} k_i u(t + \Delta t) - \sum_{i=1}^{n} k_i z_i$$

After substitution of Equations (6) and (7) into Equation (5), it is rewritten as a discrete equation at every time step, as follows:

$$R_2 u(t + \Delta t) + R_3 \frac{u(t + \Delta t)}{1 - \mu u(t + \Delta t)} + K_c u(t + \Delta t) + k_d e^{\frac{u(t+\Delta t)\pi}{2d}} = R_1 \tag{8}$$

where the detailed parameters are as follows.

$$R_1 = a(t + \Delta t) + m\left[\frac{1}{\lambda\Delta t^2}u(t) + \frac{1}{\lambda\Delta t}\dot{u}(t) + \frac{1}{(2\lambda - 1)}\ddot{u}(t)\right]$$

$$- b(\Delta t)^{-\alpha}\sum_{j=0}^{\frac{t}{\Delta t}-1}A_{j+1}\frac{u(t - j\Delta t + \Delta t)}{1 - \mu u(t - j\Delta t + \Delta t)} - C_k + \frac{\beta}{\lambda\Delta t}u(t) + \left(\frac{\beta}{\lambda} - 1\right)\dot{u}(t)$$

$$+ \left(\frac{\beta}{2\lambda} - 1\right)\Delta t\ddot{u}(t)$$

$$R_2 = \frac{m}{\lambda\Delta t^2}, R_3 = b(\Delta t)^{-\alpha}, K_e = \sum_{i=1}^{l} k_i + \frac{\beta c}{\lambda\Delta t}$$

If $\tau = u(t + \Delta t)$, Equation (8) is transformed into the following case.

$$G(\tau) = \left(R_2 + R_3 + K_c + \mu R_1 - \mu k_d e^{\frac{\tau\pi}{2d}}\right)\tau - (R_1 + K_c)\mu\tau^2 + k_d e^{\frac{\tau\pi}{2d}} - R_1 = 0 \tag{9}$$

The Newton-arithmetic mean method is selected to solve the roots of the nonlinear equation due to its wide use in algorithm research [32]. The approximate root is calculated by achieving convergence after multiple iterations with the initial values, and the iteration step is given as follows

$$\begin{cases} \tau_{k+1} = \tau_k - \frac{2G(\tau_k)}{G'(\tau_k) + G'\left(\tau_{k+1}^*\right)} \\ \tau_{k+1}^* = \tau_k - \frac{G(\tau_k)}{G'(\tau_k)} \end{cases} \tag{10}$$

where the following is the case.

$$G'(\tau) = R_2 + R_3 + K_c + \mu R_1 - \mu k_d e^{\frac{\tau\pi}{2d}} - \frac{k_d\pi}{2d}e^{\frac{\tau\pi}{2d}}(\mu\tau - 1) - 2(R_1 + K_c)\mu\tau$$

In order to rapidly and conveniently obtain the roots of the equation, the initial value of $\tau$ is often chosen as the displacement $u(t)$ at last time step, and the displacement $u(t + \Delta t)$ at the next time step is calculated by Equations (9) and (10). If the displacement for the next time step is obtained by Equation (10), the velocity and acceleration at the next time step can be obtained by substituting $u(t + \Delta t)$, $u(t)$, $\dot{u}(t)$, and $\ddot{u}(t)$ into Equation (6). Then, the time-dependent response of the nonlinear oscillator is achieved by solving the equation at each time step with the known initial condition in increasing orders of time.

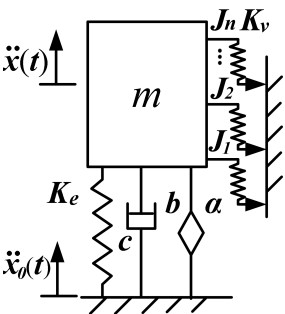

**Figure 1.** One-degree nonlinear model of isolation system.

*2.2. MFFNN for Parameter Identification*

In previous studies, the parameters of isolators can be determined by predefined variables to minimize the difference between the measured response and computed responses. Therefore, an MFFNN [33] is chosen as a framework to search the parameters of the nonlinear system in this paper. The detailed architecture of an MFFNN is constructed and illustrated in Figure 2. It is composed of three layers for parameter identification. The first layer is an input layer that transfers identified linear parameters and estimated nonlinear parameters to the current model. Since the linear parameters have been determined by the sine-sweep test, the neuron number of input layer is equal to the number of nonlinear parameters. The middle layer is used to predict the transient response of the current model with identified linear parameters and estimated nonlinear parameters by using tested input pulse $I_j$. Then, the response is transferred to the next middle layer for calculating the output values. To reduce the computational cost in the middle layer, the amplitude of the first peak $P$ and a half duration of the first pulse $t_1$ and $t_2$ are used to characterize the predicted response as shown in Figure 3. These values are used to be compared with the first peak and the half duration of the tested response in drop test, which is described as a Gaussian function:

$$h_j = exp(-\frac{\| P_j - P_j^t \|^2 + \| t_{1j} - t_{1j}^t \|^2 + \| t_{2j} - t_{2j}^t \|^2}{2\sigma_j^2}) \tag{11}$$

where $h_j$ is the output of the $j$-th neuron in the middle layer, $P_j^t$ is the amplitude of the response pulse in $j$-th test, and $t_{1j}^t$ and $t_{2j}^t$ are the half durations of the response pulse in $j$-th test. $\sigma_j$ is a standard deviation. The output layer is a linear weighted combination of the middle layer output, and its algebraic formula is as follows:

$$y = \sum_{i=1}^{m} w_i h_i - q \tag{12}$$

where $w_j$ is weights from the $j$-th neuron of the middle layer to the output layer, and $m$ is the number of neurons in the middle layer, which is equal to the number of the drop test.

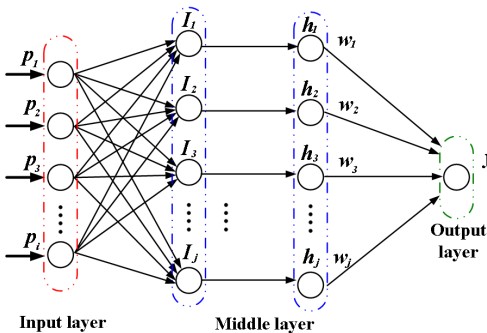

**Figure 2.** Architecture of an MFFNN.

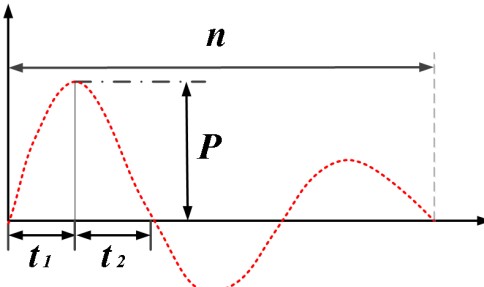

**Figure 3.** Typical impulse response of a nonlinear oscillator.

Based on the minimization process of the objective function $y$, the mid-value and deviation of each neuron and weight parameter in the hidden layer are continuously updated to support the training algorithm of the neural network via the momentum gradient descent method.

### 2.3. Identification Process

In the current model, the linear parameters include linear stiffness $K_l$ and linear damping $c$, for which its definitions are the same as these in the linear Kelvin–Voigt model. The nonlinear parameters such as $\alpha$, $b$, $d$, $k_d$, and $k_i$ are used to present the nonlinear relationship between restoring force and deformation in the dynamic equation. To decrease the number of identified parameters, the linear parameter can be determined first. Furthermore, the linear stiffness is equal to the sum of $i$ Jenkins element stiffness. It could introduce an appropriate initial condition for the identification process of the nonlinear parameters $k_i$ and may decrease the number of iteration steps required for convergence during the recognition process with the other parameters, such as a fractional order $\alpha$. Moreover, the optimized values will become closer to the real parameters of rubber isolators. Therefore, the linear modal frequency and damping of the isolation system are first identified by determining the amplitude and half-power bandwidth of the peak in the measured frequency range by a sine-sweep test. They are also used to calculate the linear stiffness and damping of rubber isolators according to the mechanical relationship between the isolator and the isolation system. The entire process of identification is described as follows:

(1) Linear stiffness and constant damping are calculated by measured stationary responses of the isolation system by the sine-sweep test.
(2) Several drop tests are carried out with a varied amplitude and delayed time of the input pulses, and the response pulses are measured in the drop test.
(3) The tested input pulses and recognized linear parameters of the rubber isolators are substituted with the other coefficients of the model into the input layer of the MFFNN framework. The response pulses are calculated by the analytical method for comparison of the amplitudes and durations between the measured and simulated acceleration response in the middle layer.

(4)   The parameters of the rubber isolator are identified by minimizing the sum of the absolute values originating from every neural path, which can be achieved by the reduced-space sequential quadratic programming (SQP) method [34].

If the optimization results are obtained in the permitted number of iterations of the identification process above, the corresponding parameters of the rubber isolator in the system are determined and exported to the presented nonlinear model. If optimization is suspended after the maximum number of iterations, the initial parameters are reset and the identification process is re-executed.

## 3. Tests System and Identification Results

### 3.1. Example of the Isolation System

High-damping silicone rubber was selected to produce a T-shaped rubber isolator. The detailed geometric parameters and the real production of the isolator are shown in Figure 4 and Table 1.

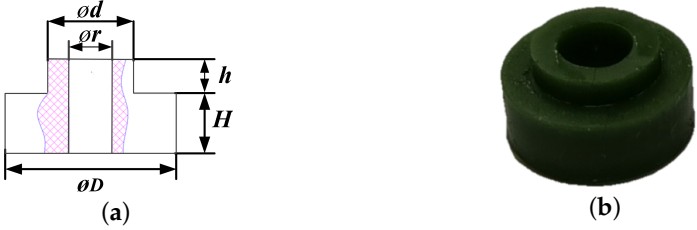

(a)                  (b)

**Figure 4.** T-shaped rubber isolator. (**a**) Sketch; (**b**) Real production

The isolation system used in the present research study is composed of equipment and four pairs of rubber isolators. Each paired rubber isolator is used to connect the equipment and basement by screwing bolts to the basement by using holes in the isolator and lug. To decrease the other directional coupling responses in the system, the installation position of the rubber isolator is located near the centroid of the equipment. The connection between the equipment and the basement is detailed in Figure 5. The stiffness of the rubber material changes significantly with compression. To keep the stiffness of four isolators as a constant value, 1.5 N/m torque is added on every bolt with a bolt tightening wrench in the installation. Due to the fact that the parameters of the four pairs of rubber isolators are equal, four of the same parallel elements are combined into a one-unit element via the linear superposition of elements with identical parameters. Then, the isolation system is simplified as a one-degree oscillator, and its model could use four-time parameters for each rubber isolator.

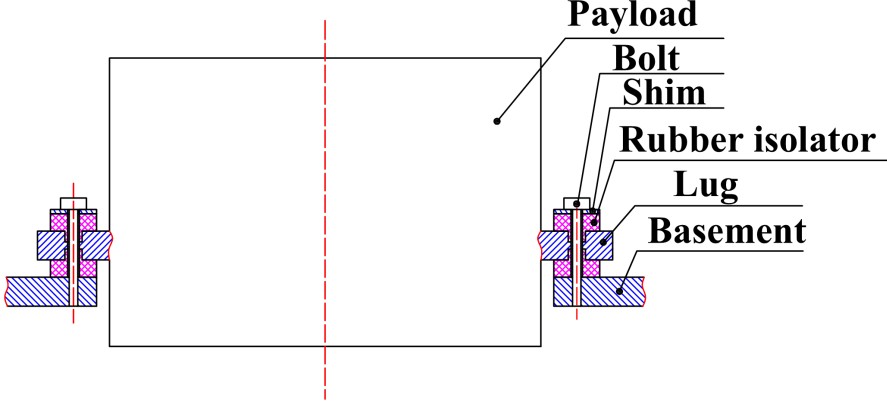

**Figure 5.** Stretch of the isolation system.

**Table 1.** Parameters of the isolation system.

| Parameter | Weight of Equipment (kg) | Rubber Isolator (mm) | | | | |
|:---:|:---:|:---:|:---:|:---:|:---:|:---:|
| | | *h* | *H* | *d* | *D* | *r* |
| Value | 1.2 | 3 | 5 | 8 | 12 | 4.2 |

*3.2. Descriptions of the Test System*

3.2.1. Sine-Sweep Test

In the identification process, the linear parameters are initially identified by a sine-sweep test. The test system consists of a driver unit and signal collection unit, which are, respectively, used to excite the system with a determined signal and to measure the acceleration responses of the system. The driver unit contained a shaker and an amplifier, and the signal collection unit included an accelerometer and a signal analyzer. The accelerometer are glued on the surface of the underwater vehicle shell and electronic equipment to pick up acceleration responses. The cylindrical shell is used to simulate the hull of a AUV and is connected to the shaker by a clamp. Then, the desired vibration generated by the shaker is transferred to the cylindrical shell. The photo of the actual experiment is shown in Figure 6. In the sine-sweep test, the input amplitude of harmonic vibration is set as 1 m/s$^2$, and the excitation frequency range is selected as 20–1000 Hz. To obtain ideal measurement results, the signal sampling frequency is chosen as 10,000 Hz, and the sweep rate was limited to 60 Hz/min.

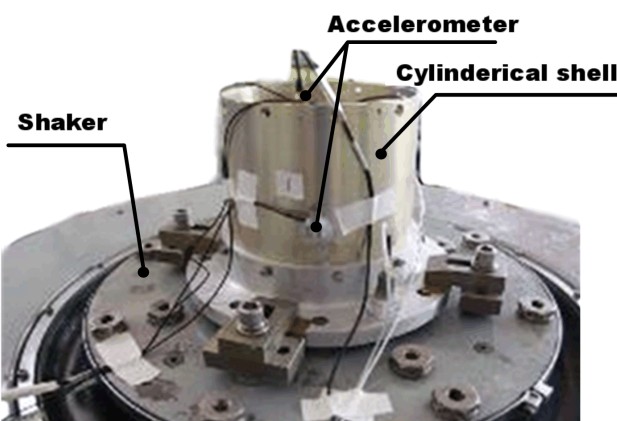

**Figure 6.** Image of the sine-sweep test system (accelerometer shaker).

3.2.2. Drop Test

The composition of the drop test system is illustrated in Figure 7. Two accelerometers are connected to the equipment and cylindrical shell of the underwater vehicle to measure the input pulse and acceleration response. The control cabinet is used to collect the acceleration signals and to analyze the shock response. Drop test equipment is used to simulate varied half-sine shocks by releasing the system at a certain height and dropping it on different buffer structures. The parameters of input pulses are controlled by the sets in the control cabinet. Furthermore, an image of the real drop test system is exhibited in Figure 8.

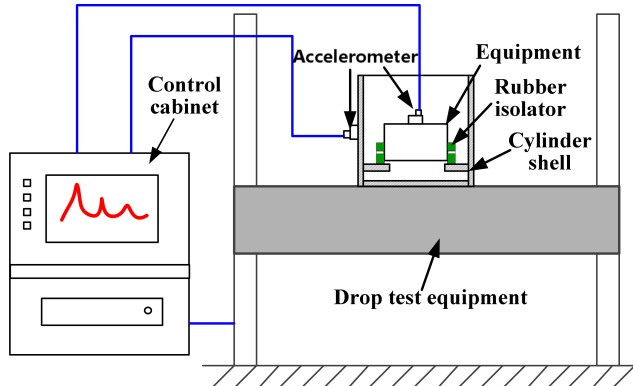

**Figure 7.** Composition diagram of the drop test system.

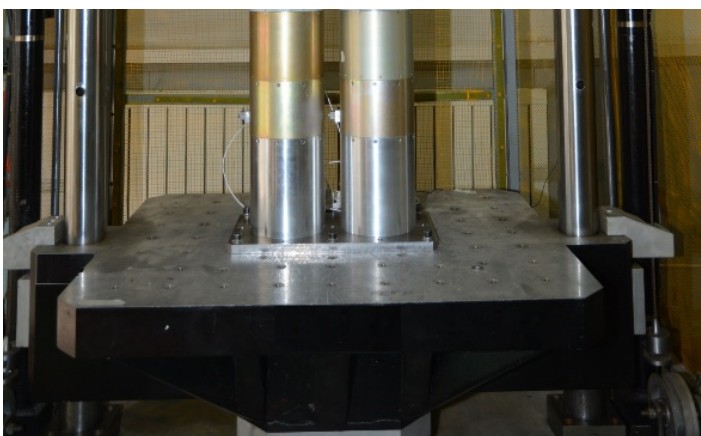

**Figure 8.** Image of the drop test system.

To obtain the detailed parameters of the rubber isolator, nine drop tests with different amplitudes and durations of the input pulse were planned and conducted with a high sampling frequency of $1 \times 10^5$ Hz to improve measurement precision. The parameters of the drop tests are listed in Table 2 and numbered from 1 to 9. By considering the cylinder shell as 2 kg, the total weight of the isolation system is 3.2 kg. The impulses added to the system are calculated by using equation below:

$$I_t = \frac{2P(t_1 + t_2)m}{\pi} \tag{13}$$

where $m$ is the total weight of the system. The values of these impulses are presented in Table 2.

**Table 2.** Parameters of nine drop tests.

| Case Number | Peak Acceleration of Input Pulse (g) | Durations of Input Pulse (ms) | Impulse (Ns) |
|:---:|:---:|:---:|:---:|
| 1 | 34.2 | 9.5 | 6.4 |
| 2 | 50.1 | 8.9 | 8.7 |
| 3 | 116.1 | 3 | 6.8 |
| 4 | 103.2 | 5.5 | 11.1 |
| 5 | 103.1 | 5 | 10.1 |
| 6 | 115.1 | 3.3 | 7.4 |
| 7 | 203.3 | 8.1 | 32.3 |
| 8 | 189.9 | 8.8 | 32.8 |
| 9 | 92.4 | 8.9 | 16.1 |

### 3.3. Identification Results

#### 3.3.1. Identification of Linear Parameters

The acceleration response of the equipment in the isolation system is measured and analyzed, which is presented in Figure 9. The rubber isolator is used to decrease vibration at low frequency, and the modal frequency of the isolation system is designed below 300 Hz. It can be obviously observed that three peaks locate between 100 Hz and 300 Hz. They may relate to the modal frequencies of the isolation system or the installed cylinder shell. Based on the finite element model of the cylinder shell with one end clamped, its first and second modal frequency are calculated as 181 Hz and 235 Hz. Therefore, the first peak of the curve presents the mode of the isolation system. The peak relevant frequency is considered as the modal frequency, and the half-power bandwidth can be used to calculate the linear damping. As exhibited in Figure 9, the modal frequency was verified as 105 Hz, and the loss factor was 0.11. According to the expressions of the modal frequency and loss factor in Equations (14) and (15), it is convenient to preliminarily estimate the linear stiffness and constant damping as $5.2 \times 10^5$ N/m and 89.4 Ns/m. They can be used as the initial parameters in the identification process.

$$\begin{cases} \omega = 2\pi f_r = \sqrt{\frac{K_l}{m}} \\ \eta = \frac{c}{\sqrt{mK_l}} \end{cases} \tag{14}$$

$$\begin{cases} K_l = m(2\pi f_r)^2 \\ c = \eta\sqrt{mK_l} \end{cases} \tag{15}$$

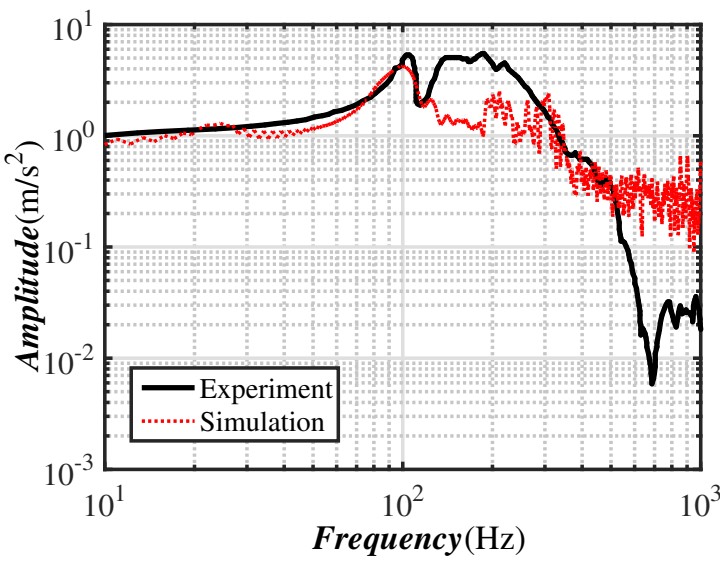

**Figure 9.** Response of the isolation system during the frequency sweep test.

#### 3.3.2. Identification of Nonlinear Parameters

In this study, the number of Jenkins elements in the Iwan model is set to tnree, which is equal to the values of $k_i$ and $z_i$ in Appendix A.1. Based on the linear stiffness and constant damping identified by the sine-sweep test, the initial parameters of the isolator are determined and substituted into the MFFNN. Then, acceleration responses are calculated with tested input pulse and combined with the results of the first eight drop tests in MFFNN. The differences between the amplitudes and durations of the acceleration response in the simulation and experiment are chosen as the objective functions in the optimization process. During the searching process, the parameters of the isolation system are updated between the upper and lower limits at each iteration and approach the actual values along the downward gradient direction of the objective function. For the equipment supported by four rubber isolator units, the parameters of the rubber isolator are equal to a quarter of

those of the entire isolation system. Therefore, the initial parameters of the isolator system estimated by the linear system, and the final parameters determined after 40 iterations are converted to the parameters of the rubber isolator, as listed in Table 3. The ranges of the isolator parameters used in the optimization process are also reported in Table 3.

**Table 3.** Initial and final parameters of the rubber isolator in the identification process.

| Parameter | $b$ | $a$ | $k_1$ (N/m) | $k_2$ (N/m) | $z_1$ (mm) | $z_2$ (mm) | $z_3$ (mm) | $c$ | $k_d$ (N/m) | $d$ (mm) |
|---|---|---|---|---|---|---|---|---|---|---|
| Lower limit | 2500 | 0.1 | $0.025k_l$ | $0.025k_l$ | 0.02 | 0.1 | 0.2 | 15 | 1 | 0.5 |
| Upper limit | 4000 | 0.5 | $0.115k_l$ | $0.115k_l$ | 0.2 | 0.5 | 1 | 75 | 10 | 4 |
| Initial | 3575 | 0.21 | $0.05k_l$ | $0.073k_l$ | 0.05 | 0.2 | 0.3 | 22.4 | 3 | 2 |
| Final | 2533 | 0.269 | $0.11k_l$ | $0.112k_l$ | 0.06 | 0.12 | 0.28 | 30.7 | 1.5 | 1.1 |

### 3.3.3. Reconstruction of Nonlinear Transient Response

The final parameters of the isolation system are substituted into the current nonlinear model, and the acceleration responses of the isolation system are predicted with the input pulses of the first eight drop tests. The calculated and measured responses are compared and shown in Figure 10. Although the amplitudes and duration of the simulated and measured responses of the system present few differences in some cases, the attenuation characteristics and trends of the curves agree with each other in Figure 10.

Additionally, the input pulse of the ninth drop test, which was not used in the MFFNN, was substituted to the current model with the identified parameters. The acceleration response of the isolation system is calculated and compared with the measured result of the ninth drop test, as presented in Figure 11. It is shown that the calculated and measured results of the acceleration response are almost the same.

The steady-state response of the isolation system against vibrational input is also important. The time-dependent input signal with 1 s duration and 1 m/s$^2$ amplitude, including serials sinusoidal signals from 10 Hz to 1000 Hz, is added into the current model with identified parameters. The time-dependent response is calculated and transferred to the frequency-dependent response by using the Fourier transform method. The response is piloted with test result in Figure 9. It is shown that the calculated and measured results of the acceleration response are almost the same near the modal frequency of the isolation system.

By examining the comparisons above, it is proved that the identified parameters of the rubber isolator are highly precise and could be adopted to correctly predict the nonlinear behavior of the isolation system, although the amplitude and duration of the shock excitation change in a wide range. Accordingly, the accuracy of the proposed model and identification method is verified.

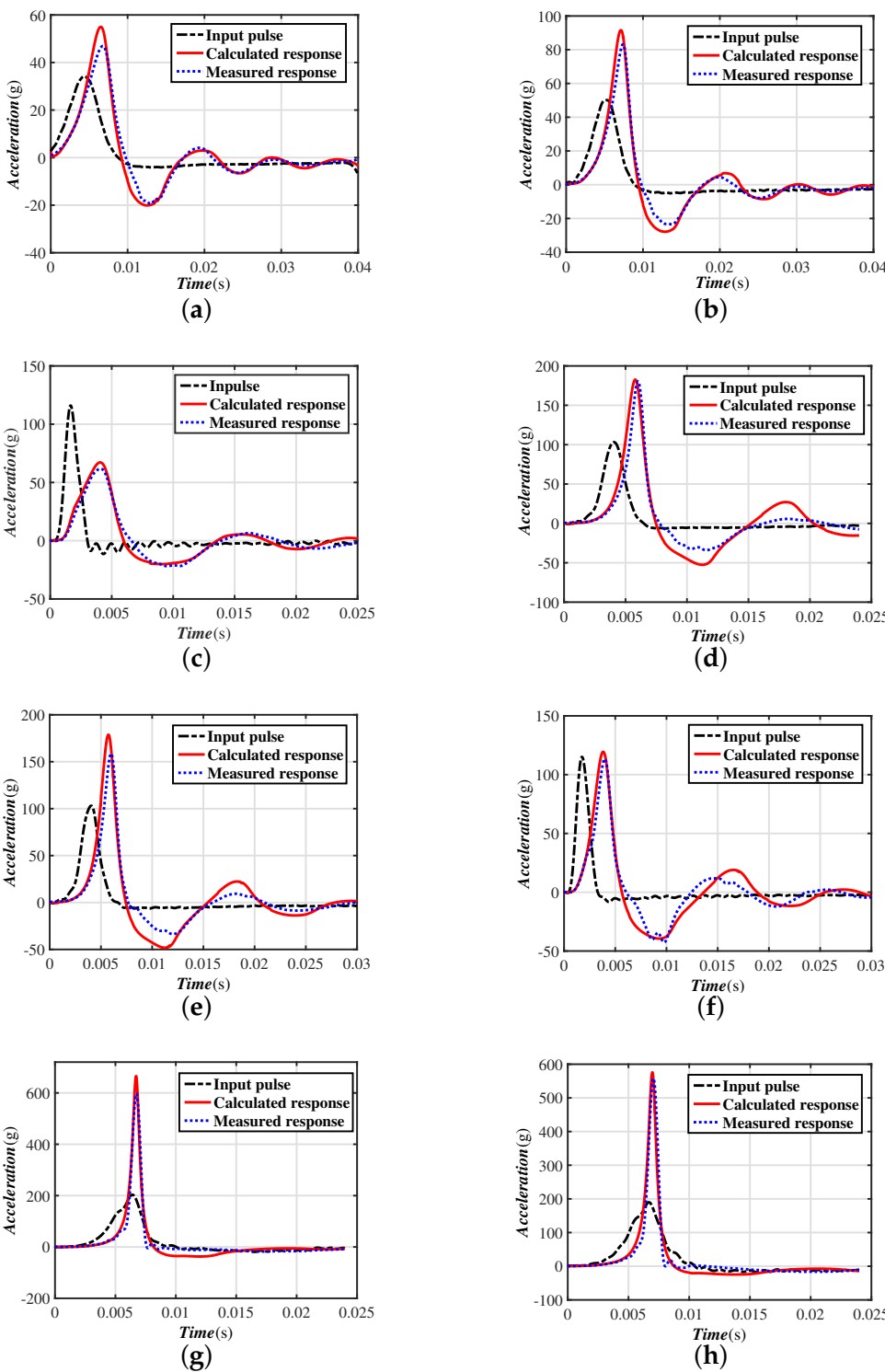

**Figure 10.** Reconstructed and measured responses of the rubber isolator in the first eight cases. (**a**) Case 1 (34.2 g, 9.5 ms); (**b**) Case 2 (50.1 g, 8.9 ms); (**c**) Case 3 (116.1 g, 3.1 ms); (**d**) Case 4 (103.2 g, 5.5 ms); (**e**) Case 5 (103.1 g, 5.0 ms); (**f**) Case 6 (115.1 g, 3.3 ms); (**g**) Case 7 (203.3 g, 8.1 ms); (**h**) Case 8 (189.9 g, 8.8 ms).

### 3.3.4. Error Analysis of the Parameter Identification

In order to provide a detailed error analysis of the simulation and test results, the calculated and measured acceleration peaks and durations of responses are selected, and their differences are reported in Table 4. It is revealed that, of all nine cases, the maximum error between the measured and calculated amplitudes reached 17.1%. While that between

the measured and calculated durations approached 10.6%. In general, the amplitude error is greater than that of the duration error. This is primarily caused by the following two reasons:

(1) Although a high sampling frequency is configured in the drop tests, it is more difficult to measure the real peak of the pulse in a short duration in which the real highest amplitude can be easily missed within a limited time interval, while this may have a lesser effect on the delay time, as the duration is sufficiently long than compared to the sampling time during the tests. Furthermore, although the rubber isolators used in this work are located near the centroid of the equipment to reduce the acceleration responses in other directions, coupling vibrations in other directions are unavoidable in real systems. This could have partially contributed to the difference between the calculated and measured acceleration responses of the system and introduced errors to the measurement of the real amplitude and duration.

(2) The effect of the attenuation properties on the response amplitude is exponential, whereas the effect on the delayed response time exhibits a mode of the root mean square. Therefore, the uncertainty in the damping parameters has a great influence on the prediction accuracy of the acceleration amplitude, while it has little influence on the prediction accuracy of the duration. This may be the key to why the proposed model predicts the delayed response time more precisely than the amplitude.

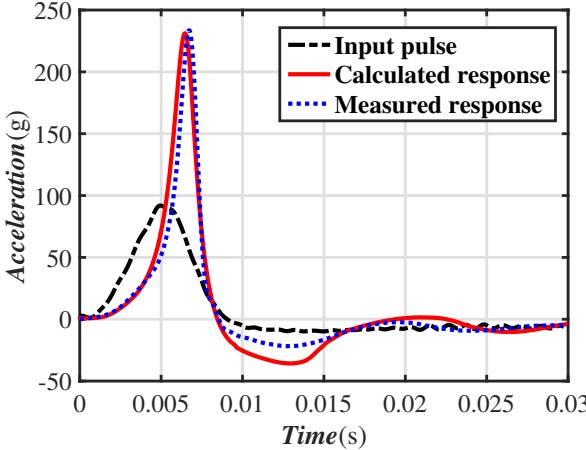

**Figure 11.** Simulated and measured responses of the rubber isolator in the ninth case.

**Table 4.** Amplitudes and durations of pulses obtained by calculations and experiments.

| Case Number | Peak Acceleration of Response Pulse (g) | | | Duration of Response Pulse (ms) | | |
|---|---|---|---|---|---|---|
| | Experiment | Calculation | Errors% | Experiment | Calculation | Errors% |
| 1 | 46.9 | 54.9 | 17.1 | 9.8 | 9.2 | 6.1 |
| 2 | 83.2 | 91.6 | 10.1 | 9.8 | 9.4 | 4.1 |
| 3 | 61.3 | 67.1 | 9.4 | 6.6 | 5.9 | 10.6 |
| 4 | 180.3 | 182.6 | 1.3 | 6.7 | 6.5 | 3 |
| 5 | 158.6 | 178.8 | 12.7 | 6.6 | 6.4 | 3 |
| 6 | 113.4 | 119.4 | 5.3 | 5.9 | 5.7 | 3.4 |
| 7 | 596.5 | 665.5 | 11.6 | 4.2 | 4.1 | 2.4 |
| 8 | 558.1 | 575.9 | 3.2 | 5.7 | 5.8 | 1.8 |
| 9 | 233.7 | 220.1 | 5.8 | 6.9 | 6.8 | 1.4 |

To analyze the effect of linear stiffness and damping on the reconstructed responses, case two is selected as the example. The linear stiffness is varied from 0.5 to 1.5 times its identified value. The linear damping is changed from 0.5 to 1.5 times of the identified linear damping. They are submitted with the identified nonlinear parameters to the current model with input pulses of case twp. Then, the pulse responses of the isolation system are calculated. Their peaks and duration are picked up and used to calculate errors between

them and test results of case two. The errors of amplitude and duration are plotted with the ratio of linear stiffness and damping, which are shown in Figure 12. It is revealed that the variances of linear stiffness and damping induce great influences on the amplitude of the reconstructed response and change the errors of amplitude between reconstructed and tested results, while the duration of the reconstructed responses is insensitive to the variance of linear stiffness and damping. Therefore, the accurate identification values of isolator linear parameters are useful for reconstructing shock response accurately, especially for its amplitude. The errors between the calculation and experiment in Table 4 are also related to the error of linear parameter identification.

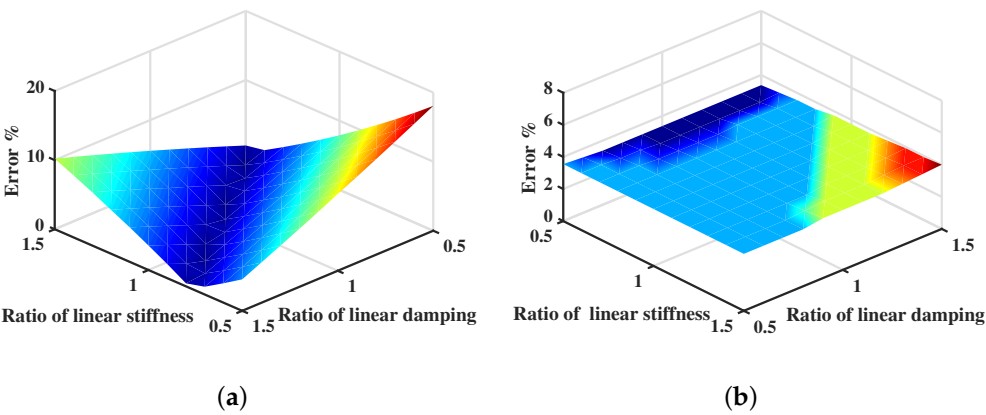

(**a**)                                     (**b**)

**Figure 12.** Effects of linear stiffness and damping on errors between reconstructed results and tested results. (**a**) Amplitude; (**b**) Duration.

*3.4. Attenuation Mode Analysis of the Rubber Isolator with Identified Parameters*

To demonstrate the attenuation principle of the rubber isolator excited by half-sine pulse with different parameters, Equation (16) is applied to calculate the dissipative energy derived from friction stiffness, fractional, and linear damping, as reported in Table 5:

$$W = \oint F dx \tag{16}$$

where $F$ is the restoring force of different damping, and $W$ is the relevant attenuation energy. The ratios between the dissipative energy of different parts and the total dissipative energy are plotted with the relevant impulses for the nine cases in Figure 13. Considering Table 5 and Figure 13, it can be concluded that the attenuation energy of friction stiffness is the main component of the total dissipative energy in case one, while the impulse is the lowest value in all cases. With the increase in impulse, the attenuation energies, originating from friction stiffness and linear damping, become close to each other in case three. With the gradual increase in impulse in cases two, four, five, six, and nine, the dissipative energy of linear damping has a dominant effect on the total energy loss. Finally, the input impulse reaches the highest level in Cases 7 and 8, and the attenuation energy of fractional damping consumes more than 50% of the total energy. In summary, with the increase in the impulse, the greatest attenuation energy unit is changed from friction stiffness to linear damping and then finally to fractional damping. Furthermore, it can be predicted that an inversion point of the main dissipative energy source could exist when the impulse amplitude increases continuously. However, this conclusion must be proven in future research studies via a large number of simulations and tests.

**Table 5.** Dissipation energies of different restoring forces in the nine drop tests.

| Case Number | Dissipation Energy (J) | | | Total Energy (J) |
|:---:|:---:|:---:|:---:|:---:|
| | Friction Stiffness | Fractional Damping | Linear Damping | |
| 1 | 0.188 | 0.073 | 0.167 | 0.428 |
| 2 | 0.290 | 0.170 | 0.364 | 0.825 |
| 3 | 0.483 | 0.110 | 0.487 | 1.080 |
| 4 | 0.509 | 0.477 | 0.970 | 1.956 |
| 5 | 0.498 | 0.461 | 0.934 | 1.893 |
| 6 | 0.405 | 0.311 | 0.681 | 1.397 |
| 7 | 0.463 | 1.441 | 1.093 | 2.997 |
| 8 | 0.379 | 1.187 | 0.798 | 2.364 |
| 9 | 0.423 | 0.516 | 0.656 | 1.595 |

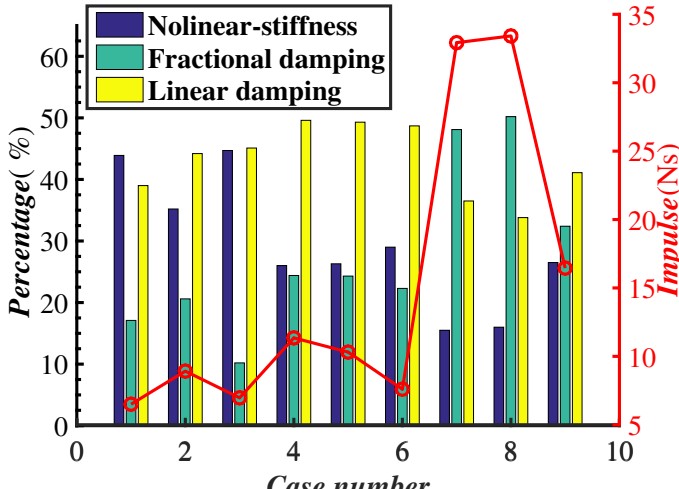

**Figure 13.** Relationship between the dissipation energy and input impulse for different cases.

## 4. Conclusions

This article proposes a nonlinear model of rubber isolators for simulating their behavior under shock pulses. The model is composed of four components to describe the amplitude and frequency-dependent effects of the rubber material. Then, the nonlinear dynamic equation of the isolation system is constructed. It is solved by the Newmark method and Newton-Arithmetic mean method in order to obtain the transient response of the isolator.

Based on the current model, the architecture of an MFFNN is constructed for determining parameters of the isolator with measured and simulated responses. To accelerate the iteration process of the MFFNN, the linear parameters are determined by the sine-sweep test and used as initial values for the MFFNN. Then, the parameter identification process of the rubber isolator under shock excitation is proposed with sine-sweep and drop test results.

Then, a silicon T-shape isolator is selected as a sample case. Its parameters are identified by the current method with sine-sweep and drop test results. The transient responses of the isolator system induced by different shock pulses are reconstructed by the identified parameters, which show good agreements with the tested responses. It is proved that the identified parameters of the rubber isolator are highly precise and could be adopted to correctly predict the nonlinear behavior of the isolator system.

Finally, the errors between the reconstructed responses and measured responses are analyzed. It was found that the amplitude error is greater than that of the duration error due to difficult measurement and estimation perturbations of the peak amplitude. The energy

attenuation mode of the rubber isolator is studied with the identification parameters and test results. It was found that the main portion of consumed attenuation energy changed from friction stiffness to linear damping and finally transformed to fractional damping with the increase in impulse amplitude.

**Author Contributions:** methodology, H.X. and R.W.; validation, C.X. and P.Y.; formal analysis, J.Z.; writing—original draft preparation, H.X.; writing—review and editing, J.B. All authors have read and agreed to the published version of the manuscript.

**Funding:** This work is supported by grants from National Natural Science Foundation of China (NSFC) (grant number 12002283) and Key Laboratory of Aerodynamic Noise Control (grant number ANCL20190307).

**Institutional Review Board Statement:** Not applicable.

**Informed Consent Statement:** Not applicable.

**Data Availability Statement:** Not applicable.

**Conflicts of Interest:** The authors declare no conflict of interest.

## Appendix A. Restoring Force Models of the Rubber Isolator

### Appendix A.1. Frictional Force Model

In order to describe the viscoelastic behavior of the rubber material in an isolator, the Iwan [25] model is used to construct networks of springs and frictional sliders to simulate the hysteresis effect of rubber material. In order to reduce the computational cost, the Jenkins element is characterized by the variation of the elastic stiffness and Coulomb slider. The parallel-series model is then modified to a revised model, as shown in Figure A1. To introduce the variation of the elastic stiffness and Coulomb slider, the integrated formulation of the Iwan model is changed to a dispersed formulation, as given by Equation (A1). Upon initial loading, the force-deflection relationship for the Jenkins element of the system will have the following form.

$$\begin{cases} f_i = k_i x & \dot{x} > 0, 0 \le x \le f_i/k_i \\ f_i = f_i^* & \dot{x} > 0, x > f_i/k_i \end{cases} \tag{A1}$$

If the direction of loading is reversed after the Jenkins element has yielded, the force-deflection relationship will become the following:

$$\begin{cases} f_i = k_i x - (k_i L_i - f_i^*) & \dot{x} < 0, L_i - 2f_i^*/k_i \le x \le L_i \\ f_i = -f_i^* & \dot{x} < 0, x \le L_i - 2f_i^*/k_i \end{cases} \tag{A2}$$

where $L_i$ is the maximum deflection of the $i$-th Jenkins element, and $k_i$ and $f_i^*$ are, respectively, the elastic stiffness and Coulomb slider force of $i$-th Jenkins element.

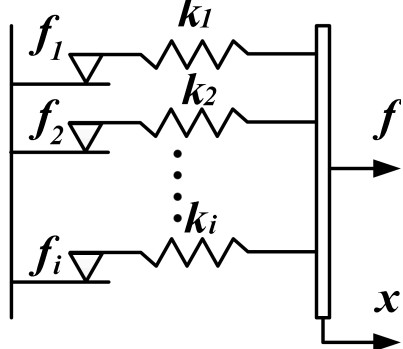

**Figure A1.** Modified Iwan spring-slider model.

The relationship between the force and deformation of the modified Iwan model can be described as the sum of all Jenkins elements as follows:

$$F_f[x(t),(z_i)] = \sum_{i=1}^{N} \left\{ \begin{array}{ll} k_i(x(t) - z_i), & |x(t) - z_i| < L_i \\ f_i^* \operatorname{sgn}(x(t)), & \text{else} \end{array} \right. \tag{A3}$$

and the following is the case:

$$K_l = \sum_{i=1}^{N} k_i$$

where $z_i$ is the displacement of the Coulomb slider in the $i$-th Jenkins element, and $K_l$ is the linear stiffness of the oscillator.

*Appendix A.2. Hyperplastic Force Model*

In order to simulate the hyper-elastic phenomenon of rubber, the restoring force $F_e(t)$ is assumed as exponential elasticity. This force-deflection characteristic is given as follows:

$$F_e(t) = k_d e^{\frac{\pi x(t)}{2d}} \tag{A4}$$

where $k_d$ is the initial exponential stiffness of rubber, and $d$ is the maximum deflection of the infinial force.

By using appropriate parameters in the model of amplitude-dependent force, the relationship between the force and deformation is translated and plotted with previously reported ramp test results of silicone rubber plane compression [24] in Figure A2. The figure reveals that the simulated results of the proposed model match well with test results, thereby proving the accuracy of the proposed model.

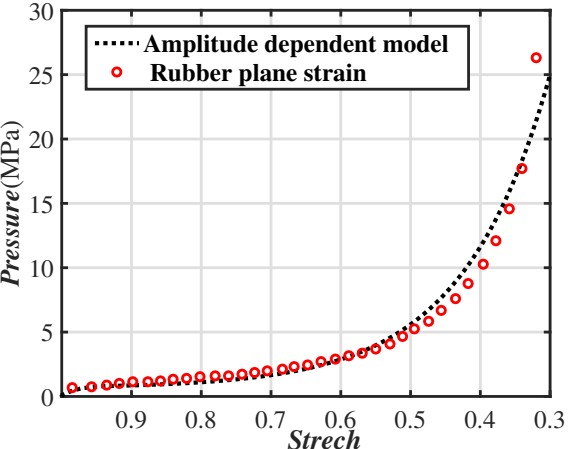

**Figure A2.** Simulated behavior and ramp test results of silicone rubber plane compression [24].

*Appendix A.3. Fractional-Order Model*

In order to obtain a good description of the frequency dependence effect of viscoelastic materials, time derivatives of non-integer order are usually used to illustrate the constitutive relationships to reduce the required number of parameters. Thus, the partial frequency dependence effect of the proposed model is realized as a fractional derivative order where the viscous forces are modeled by a time-discrete equation as follows [20]:

$$F_{fr}[x(t)] = bD^{\alpha} \frac{x(t)}{1 - \mu x(t)} \approx b(\Delta t)^{-\alpha} \sum_{j=0}^{\frac{t}{\Delta t} - 1} A_{j+1} \frac{x(t - j\Delta t)}{1 - \mu x(t - j\Delta t)} \tag{A5}$$

where $A_{j+1}$ is listed as below:

$$A_{j+1} = \frac{\Gamma(j-\alpha)}{\Gamma(-\alpha)\Gamma(j+1)} \tag{A6}$$

In Equation (A5), $b$ is a proportionality constant, and $\alpha$ is the time derivative order. When a modeled oscillator uses a spring-pot in parallel with a spring and Coulomb slider, the order of time derivative $\alpha$ is normally located between 0 and 1. Equation (A5) reveals that the previous history of the displacement is considered by summing over the weighted $x(t-j\Delta t)$. Moreover, the influence of the displacement history on the fractional viscoelastic force is gradually degraded by using $A_{j+1}$ as a weighting function, which indicates that displacement history is truncated and partially takes part in the equation.

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
