# Peer review of "A Nonlinear Model and Parameter Identification Method for Rubber Isolators under Shock Excitation in Underwater Vehicles"

_jmse, doi:10.3390/jmse9111282_

Round 1
Reviewer 1 Report
The reviewer believes that this research is valuable and exciting.
Main comments:
Introduction
- The authors proposed the sine-sweep and dropped tests as the initial value to accelerate iteration in the MFFNN process. The reviewer can not find any previous related papers of the sine-sweep and drop tests. The reviewer believes that this paper is not the first publication for the experimental dropped test.
Section 2.1 - Nonlinear dynamic model of an isolation system
- This paper proposed the nonlinear dynamic model for underwater vehicles. The reviewer can not find the added mass for the floating body in this model. If it is neglected for the simplification model, please let us know the reason.
Section 2.2 – MFFNN for parameter identification
- The paper showed that the sine sweep test is conducted to determine the linear parameter and used as an initial value. How is the mechanism of dropped tests combination?
Section 3.1 Example of the isolation system
- Could the authors demonstrate what the linear and nonlinear parameters are?
Section 3.3 – Identification results:
- 11 presented the simulated and measured response of the rubber isolator. However, the comparison with and without the sine-sweep and dropped tests as the initial value is not presented. How is the improvement when the sine sweep and drop test are planned with MFFNN to build the parameter identification process of rubber isolators? Then, the reviewer can not find the advantages of the author’s method while the sine-sweep and dropped tests are added.
Minor comments:
- Eq. 4&5, notation for dot derivative is shifted in pdf file (page 4)

Author Response
Dear editor:
Thanks for your help and the comments of reviewers on my paper. We have studied the valuable comments of reviewers and tried our best to revise the manuscript. Firstly, the grammar of the whole paper is checked and corrected . Then, according to the reviewer's comments, the whole paper is revised. Several equations and figures are added to the revised paper. The equations and figures are renumbered. All the revised contents are marked up using the “Track Changes” function. The detailed responses and revised contents to the reviewer's comments are listed as follows:
- Response to the comments of reviewer 1
Comment 1:Section 2.1 - Nonlinear dynamic model of an isolation system
- This paper proposed the nonlinear dynamic model for underwater vehicles. The reviewer can not
find the added mass for the floating body in this model. If it is neglected for the simplification
model, please let us know the reason.
Response 1: Thanks for the comment of the reviewer. The linear parameters are identified firstly, it could decrease the number of identified parameters in the MFFNN and reduce the time cost of the identification process. While usage of the data in sine sweep test and drop test in MFFNN has no improvement for the method itself. To give a more accurate introduction of the identification method and process, the descriptions are revised in the introduction section of the revised paper as follow:
The drop tests are planed to obtain the measurement data which is used as baselines of the comparisons to search parameters in MFFNN. In consideration of numerous linear and nonlinear identification parameters, a sine sweep test is conducted to determine the linear parameters and used as the initial value to accelerate the parameter searching process and improve optimization in MFFNN efficiency.
Comment 2: Section 2.2 – MFFNN for parameter identification
- The paper showed that the sine sweep test is conducted to determine the linear parameter and used as an initial value. How is the mechanism of dropped tests combination?
Response 2: Thanks for the comment of the reviewer, the combination between the sine-sweep test and drop test is explained as: The first layer is an input layer that transfers identified linear parameters and estimated nonlinear parameters to the current model. Since the linear parameters have been determined by the sweep-sine test, the neuron number of the input layer is equal to the number of nonlinear parameters. The middle layer is used to predict the transient response of the current model with identified linear parameters and estimated nonlinear parameters, combing with tested input pulse Ij. These contents are added in section 2.2 of the revised paper.
Comment 3: Section 3.1 Example of the isolation system
- Could the authors demonstrate what the linear and nonlinear parameters are?
Response 3: The descriptions of the linear and nonlinear parameters are added in section 2.3. And the relationship between linear stiffness and Jenkins element stiffness is presented for explaining why the linear stiffness can work as inertial values for the nonlinear parameter identification. All these contents are clearly presented in section 2.3 of the revised paper.
Comment 4: Section 3.3 – Identification results:
- Fig. 11 presented the simulated and measured response of the rubber isolator. However, the
comparison with and without the sine-sweep and dropped tests as the initial value is not
presented. How is the improvement when the sine sweep and drop test are planned with MFFNN
to build the parameter identification process of rubber isolators? Then, the reviewer can not find
the advantages of the author’s method while the sine-sweep and dropped tests are added.
Response 4: Effects of the linear parameters on the predicted impact response of the isolation system are discussed in section 3.3.4 of the revised paper. It is revealed that the prediction accuracy of the impact response is related to the accuracy of the linear parameters, especially for the amplitude of the responses. Therefore, the sine-sweep test determines the linear parameter firstly before nonlinear parameter identification. The number of identified parameters in the MFFNN could reduce and the accuracy of the identified results could be ensured. These contents are presented in the revised paper.
Comment 5: Minor comments:
- Eq. 4&5, notation for dot derivative is shifted in pdf file (page 4)
Response 5:
Thanks for your suggestion, I have shifted the notation for dot derivative in Eq. 4&5 and marked them up using the “Track Changes” function.

Reviewer 2 Report
Please find attached file.

Author Response
Dear editor:
Thanks for your help and the comments of reviewers on my paper. We have studied the valuable comments of reviewers and tried our best to revise the manuscript. Firstly, the grammar of the whole paper is checked and corrected. Then, according to the reviewer's comments, the whole paper is revised. Several equations and figures are added to the revised paper. The equations and figures are renumbered. All the revised contents are marked up using the “Track Changes” function. The detailed responses and revised contents to the reviewer's comments are listed as follows:
- Response to the comments of reviewer 2
Comment 1: The paper deals with rubber modeling to accurately predict the response when a rubber-supported device (payload) receives a shock impulse. It is shown that accurate prediction for impulse response is possible by the proposed method. In an actual situation, the steady-state response against vibrational input, such as random or sinusoidal input, is also considered necessary. How about the estimation performance of the response for swept sinusoidal input shown in Fig. 9? Could you add some additional results for it?
Response 1: Thanks for your good suggestion. The author also thinks that the steady-state response against vibrational input is important for the isolation system. Therefore, the reconstructed sinusoidal response of the isolation system with identified parameters is calculated and plotted with measurement results in Figure 9. It is seen that the estimated sinusoidal response agrees well with the tested results near the modal frequency of the isolation system. Considering the topic and focus area of the manuscript, the process of calculating sinusoidal response is briefly presented in section 3.3.3 of the revised paper.
Comment 2: From Fig.9, it seems that the characteristics different from the one-degree-of freedom system assumed in eq.(13) appear. E.g., there also looks like to have one resonant peak with relatively large damping in the range of 100-200Hz as a whole; and a clear anti-resonant frequency is found around 120-130Hz. There is no single peak in the response curve, and it can also be seen that the maximum response is observed around 180Hz. Could you please give an additional explanation about the reason that the natural frequency fr is 105Hz is verified from the data?
Response 2: The rubber isolator is used to decrease vibration at low frequency and the modal frequency of the isolation system is designed below 300Hz. It can be obviously observed that three peaks locate between 100Hz and 300Hz. They may relate to the modal frequencies of the isolation system or the installed cylinder shell. Based on the finite element model of the cylinder shell with one end clamped, its 1st and 2nd modal frequencies are calculated as 181Hz and 235Hz. The mode shapes of those two modes are plotted as below. Therefore, the first peak of the curve presents the mode of the isolation system. These contents are added in section 3.3.1of the revised paper.
Figure 1 1st mode shape of the installed cylinder shell
Figure 2 2nd mode shape of the installed cylinder shell
Comment 3: It is said that linear parameters are used as initial values when identifying non-linear parameters by NN; however, in the case of rubber materials, it is often difficult to make a clear judgment when identifying linear parameters, as shown in Fig. 9. If possible, please add data to explain the effect on the identification result and the prediction accuracy of the impact response when the linear parameters change.
Response 3: According to the suggestion of the reviewer, the effects of the linear parameters on the predicted impact response of the isolation system are added in section 3.3.4 of the revised paper. It is revealed that the accurate identification values of isolator linear parameters are useful for reconstructing shock response accurately, especially for its amplitude. The errors between the calculation and experiment are also related to the error of linear parameter identification. The detailed discussion is presented in section 3.3.4 of the revised paper.
Comment 4: Looking at Fig. 5, it seems that the rubber isolator is used while being compressed by bolts. Is this right? If so, in our experience, the stiffness of the rubber material changes significantly with compression, so it is not easy to keep it constant. How are you controlling the rubber rigidity of the isolator?
Response 4: Thanks for the comment of the reviewer. The stiffness of the rubber material changes significantly with compression in reality structure. To keep the stiffness of four isolators as a constant value, 1.5N/m torque is added on every bolt with a bolt tightening wrench in the installation. These contents are added in section 3.1 of the revised paper.
Comment 5: The reviewer cannot find explanations about how the Impulse in Fig. 12 is calculated. Does it correspond to A in Fig. 3 or does it depend on A and the time widths t1 and t2? Please add an explanation. The same applies to Table 2, too.
Response 5: The relationship between impulse and A, t1,t2 is described in equation 13 of the revised paper.
Comment 6: Followings are the editorial comments.
1) Is the "accelerator" that appeared in Fig. 6 the "accelerometer"? If so, The reviewer thinks it is appropriate to use "accelerometer" or "acceleration pickup."
2) Is the "shaker driver" that appeared in Fig. 6 mean the equipment to excite the test piece? If so, The reviewer thinks it is appropriate to use "shaker" or "exciter." Many readers may imagine the "shaker driver" as an "amplifier and control device for a shaker."
3) What does the "cylindrical shell" in Fig.6 represent? Is it simulating the hull of an AUV? Please add a description.
4) What does q in equation (12) represent? Is there a relationship with q in equation (2)?
5) What is v in equation (14)? Isn't it fr in equation (13)?
6) Please check the spell of English words. For example;
- L. 72: frequency rang => frequency range?
- L.108: ractional => fractional?
- L.330: infiinal force?
Response 6:
1) Thanks for your suggestion, I have changed the “accelerator” to “accelerometer” in Figure 6 and marked it up using the “Track Changes” function.
2) Thanks for your suggestion, I have changed the “shaker driver” to “shaker” in Figure 6 and marked it up using the “Track Changes” function.
3) The cylindrical shell presents the hull of an AUV. According descriptions are added in section 3.2.1 of the revised paper.
4) Because of a clerical error, the α is written as the q in the equation (2). It has been corrected in the revised paper.
5) Thanks for your suggestion, I have changed the “ν” to “fr” in Figure 6 and marked it up using the “Track Changes” function.
6) Thanks for your suggestion, I have checked the spell of English words. All the changes are marked up using the “Track Changes” function.

Reviewer 3 Report
Accept in present form
Author Response
Dear editor:
Thanks for your help and the comments of reviewers on my paper. We have studied the valuable comments of reviewers and tried our best to revise the manuscript. Firstly, the grammar of the whole paper is checked and corrected . Then, according to the reviewer's comments, the whole paper is revised. Several equations and figures are added to the revised paper. The equations and figures are renumbered. All the revised contents are marked up using the “Track Changes” function.